# Multiple Congenital Anomalies-Hypotonia-Seizures Syndrome 2 Caused by a Novel *PIGA* Variant Not Associated with a Skewed X-Inactivation Pattern

**DOI:** 10.3390/genes15060802

**Published:** 2024-06-18

**Authors:** Alba Gabaldon-Albero, Lourdes Cordon, Amparo Sempere, Laia Pedrola, Carla Martin-Grau, Silvestre Oltra, Sandra Monfort, Alfonso Caro-Llopis, Marta Dominguez-Martinez, Sara Hernandez-Muela, Monica Rosello, Carmen Orellana, Francisco Martinez

**Affiliations:** 1Translational Genetics Research Group, Instituto de Investigacion Sanitaria La Fe (IIS La Fe), 46026 Valencia, Spain; alba_gabaldon@iislafe.es (A.G.-A.); pedrola_lai@gva.es (L.P.); carla_martin@iislafe.es (C.M.-G.); oltra_jua@gva.es (S.O.); monfort_san@gva.es (S.M.); alfons_caro@iislafe.es (A.C.-L.); marta_dominguez@iislafe.es (M.D.-M.); rosello_mpi@gva.es (M.R.); orellana_car@gva.es (C.O.); 2Pediatric Neurology Unit, Hospital Universitario y Politecnico La Fe, 46026 Valencia, Spain; hernandez_sarmue@gva.es; 3Hematology Research Group, Instituto de Investigacion Sanitaria La Fe (IIS La Fe), 46026 Valencia, Spain; lourdes_cordon@iislafe.es (L.C.); sempere_amp@gva.es (A.S.); 4Centro de Investigación Biomédica en Red de Cáncer (CIBERONC), Instituto Carlos III, 20029 Madrid, Spain; 5Hematology and Hemotherapy Service, Hospital Universitario y Politecnico La Fe, 46026 Valencia, Spain; 6Genetics Unit, Hospital Universitario y Politecnico La Fe, 46026 Valencia, Spain

**Keywords:** phosphatidylinositol glycan-class A protein, epileptic encephalopathy, drug resistant epilepsy, X-linked, flow cytometry

## Abstract

Germline variants in the phosphatidylinositol glycan class A (*PIGA*) gene, which is involved in glycosylphosphatidylinositol (GPI) biosynthesis, cause multiple congenital anomalies-hypotonia-seizures syndrome 2 (MCAHS2) with X-linked recessive inheritance. The available literature has described a pattern of almost 100% X-chromosome inactivation in mothers carrying *PIGA* variants. Here, we report a male infant with MCAHS2 caused by a novel *PIGA* variant inherited from his mother, who has a non-skewed pattern of X inactivation. Phenotypic evidence supporting the pathogenicity of the variant was obtained by flow-cytometry tests. We propose that the assessment in neutrophils of the expression of GPI-anchored proteins (GPI-APs), especially CD16, should be considered in cases with variants of unknown significance with random X-inactivation in carrier mothers in order to clarify the pathogenic role of *PIGA* or other gene variants linked to the synthesis of GPI-APs.

## 1. Introduction

Germline variants in the phosphatidylinositol glycan class A (*PIGA* [OMIM 311770]) gene cause multiple congenital anomalies-hypotonia-seizures syndrome 2 (MCAHS2 [OMIM 300868]) with an X-linked recessive pattern of inheritance. PIGA is one of the seven proteins involved in the first step of glycosylphosphatidylinositol (GPI) biosynthesis [1], enabling the anchoring of proteins to the cellular surface. As a result of somatic pathogenic variants in *PIGA*, erythrocytes lose the CD55 and CD59 GPI-anchored proteins (GPI-APs), causing paroxysmal nocturnal hemoglobinuria (PNH [OMIM 300818]). In 2012, germline disease-causing variants in *PIGA* were described by Johnston et al. in a family with several male patients displaying multiple congenital malformations, epilepsy, and severely delayed psychomotor development [2]. The mechanism by which the alteration of GPI causes neurological symptoms is not completely understood, but normal function of GPI-APs is key for central nervous system (CNS) development and function. In recent years, more cases of MCAHS2 with a broad spectrum of severity and another phenotype also showing haemochromatosis (NEDEPH [OMIM 301072]) have been reported [3]. Interestingly, the available literature has described skewed X-chromosome inactivation with almost 100% prevalence in mothers carrying the *PIGA* variant [2,4,5]. The transcriptional silencing of one of the X chromosomes in XX individuals, also known as X-chromosome inactivation, is a dosage-compensation strategy that occurs in most mammals. In placental mammalsthe prevalent form is random X-chromosome inactivation, so the paternal and maternal X chromosomes have roughly the same chance of being inactivated [6]. However, preferential (skewed) inactivation of one X chromosome can occur by chance, due to a non-random choice [7], or as a result of selection for or against cells carrying one specific active or inactive X chromosome. This last mechanism occurs in individuals that carry X-linked variants associated with lethality or restricted survival and is a hallmark of carriers of a few X-linked diseases [6]. There are several X-linked disorders for which all the pathogenic variants are systematically associated with complete skewed X-inactivation, such as disorders associated with the *ATRX* gene [8] or incontinentia pigmenti [9]. In these cases, preferential X-inactivation is pathognomonic and its identification serves as a supporting functional diagnostic. Other X-linked conditions are frequently, but not always, associated with preferential X-inactivation [10], so the analysis of the X-inactivation pattern may be suggestive, although random X-inactivation does not exclude their involvement. For the pathogenic variants in the *PIGA* gene, it was thought that asymptomatic female carriers would show preferential X-inactivation. In fact, whenever these studies have been reported, preferential inactivation has been found in carrier females [2,4,5,11]. Thus, it could be inferred that *PIGA* variants with random inactivation in females do not have a deleterious effect and that their pathogenic role must therefore be questioned.

Here we report a male infant with MCAHS2 caused by a novel *PIGA* variant inherited from his mother, who has a non-skewed pattern of X inactivation.

## 2. Materials and Methods

### 2.1. Exome Sequencing

Patient and maternal genomic DNA was obtained from peripheral blood leukocytes using standard methods. Massive parallel sequencing for exome sequencing was performed using the SureSelect CCP17 Exome (Agilent Technologies, Santa Clara, CA, USA) and run on the Illumina NextSeq500 platform following the manufacturer’s protocol to obtain a minimum reading depth of 100X. Read alignments, variant calling, and annotations were performed in the Alissa Interpret platform (Agilent Technologies). Disease-causing genes related to neurodevelopmental disorders, and candidate genes reported in different databases were analyzed. To evaluate the clinical impact and to assess the pathogenicity of the variants, previously reported criteria were used [12,13]. The novel *PIGA* variant was confirmed by Sanger sequencing (primers and PCR conditions are available on request) and submitted to the Decipher data-base (www.deciphergenomics.org (accessed on 14 December 2023), reference number 524900).

### 2.2. Flow-Cytometry Analysis and Cell Sorting of GPI-APs-Deficient Hematopoietic Cells

To evaluate the expression of GPI-APs in neutrophils and monocytes, the highly sensitive flow-cytometry techniques recommended for diagnosis of PNH were initially used [14]. Briefly, first, 100 μL of fresh blood was stained using a reagents cocktail of five fluorochromes (Table 1). In the second step, two combinations of markers with eight different fluorochromes were employed in two different tubes (Table 1). In both protocols, the red blood cells were lysed with FACS Lysing Solution (BD Bioscience, Becton Dickinson, San Jose, CA, USA). After lysis, the samples were centrifuged, washed with PBS supplemented with 1% BSA, resuspended in 0.5–1 mL of PBS, and analyzed in a FACSCanto-II flow cytometer (Bec-ton Dickinson, San Jose, CA, USA). Several gating steps were performed to identify GPI-deficient cells: after the selection of leukocytes and singlets by forward (FSC) and side-scatter (SSC) parameters, neutrophils and monocytes were identified based on the expression of the selection markers CD15 + CD45 or CD64 + CD45, respectively. Finally, the neutrophil population was assessed according to the expression of CD16, CD24, CD157, and FLAER, while monocytes were assessed with CD14, CD157, and FLAER GPI markers. Infinicyt v2.0 software was employed for the analyses.

Cell sorting was performed in a FACSAria-III cell sorter (Becton Dickinson, San Jose, CA, USA), using a similar strategy. A total of 100 μL of sample was stained with CD15-HV450, CD45-HV500, CD157-PE, CD16-PECy7, CD20-PerCP, CD64-APC, and FLAER and lysed using FACS Lysing solution (see Table 1). In the last step, two neutrophil populations were isolated according to the expression of CD16. Cells were sorted by purity using a 100 μm nozzle until the sample ran out. We isolated 26,780 CD16+ cells and 15,352 CD16+low/−cells with efficiencies of 89.40% and 89.96%, respectively. The post-sort purity values were 89.1% and 98.1%, respectively.

### 2.3. X Inactivation Studies

The inactivation pattern of the X chromosome was studied as reported elsewhere [15] with minor modifications. Briefly, two genomic DNA samples were digested with the methylation-sensitive enzyme Hpa II or with Rsa I (as a control) (New England Biolabs, Ipswich, MA, USA). Once digested, the microsatellite genetic marker located in the promoter region of the AR gene, which is hypermethylated on the inactivated X chromosome in each cell, was characterized by PCR amplification and fragment analysis in the ABI-PRISM3130 analyzer, Thermo Fisher Scientific, Waltham, MA, USA).

## 3. Results

### 3.1. Clinical Features

The index case is a male infant born from healthy non-consanguineous parents, being the third spontaneous pregnancy of his 32-year-old mother, who had two previous miscarriages. He was born at term by uneventful vaginal delivery. At the age of two months, he had no visual fixation, severe hypotonia, and abnormal general movements. At five months, he had a generalized tonic seizure, and two months later, he was admitted for status epilepticus. No abnormalities in the internal organs or CNS were detected, and blood tests were normal, including alkaline phosphatase and iron-metabolism parameters. At 15 months, he was severely developmentally delayed, had not reached any developmental milestones, and had severe global hypotonia, chorea, dysphagia, and minor facial dysmorphism (see Figure 1). He suffered multiple daily generalized tonic seizures despite various combinations of antiseizure medications. The interictal EEG can be seen in Figure 1. The ketogenic diet, pyridoxine, and hormone therapy (methylprednisolone) were not effective either. The frequency and intensity of seizures increased, and an intravenous infusion of midazolam was necessary for control. He finally passed away at 18 months in the context of status epilepticus. 

Regarding family history, there was a male cousin of the mother, who suffered encephalopathy of unknown origin with epilepsy and severe intellectual disability, although genetic studies were not performed. No relevant phenotypic features were found in the mother, and the reported problems to date in her medical history were obesity, polycystic ovary syndrome, and gestational diabetes.

### 3.2. Identification of PIGA Variant

The novel missense hemizygous variant NM_002641.4:c.130C>G [p.Pro44Ala] was found in the *PIGA* gene. This variant has not been previously described in patients or controls (absent from gnomAD, ClinVar, etc. databases and our in-house database). It occurred at a highly conserved amino acid and was predicted to have a damaging functional effect by REVEL, MetaRNN, BayesDel, PROVEAN, CADD, etc., which supported its pathogenicity (ACMG criteria: PP3, PM1, PM2). The study using Sanger sequencing determined that the mother was a heterozygous carrier of the variant. 

### 3.3. Flow Cytometry and Cell Sorting

Analysis was performed on leucocytes from the patient, his mother, and a normal control sample. Among the maternal neutrophils, a subpopulation with a significant decrease in CD16 expression was detected. This subpopulation represented 26% of neutrophils, while 74% had normal CD16 expression. These two subpopulations were selected and sorted for further genetic testing. In the CD16-deficient clones, a moderate decrease in the expression intensity of FLAER and a less significant decrease in CD24 expression were observed, while CD157 expression was nearly normal (see Figure 2). In the patient, 100% of neutrophils displayed the same characteristics as the clone detected in his mother: the surface expression of CD16 was significantly decreased, and a moderate reduction in FLAER expression was observed. Expression of CD157 was normal, and a very slight decrease was noted for CD24 when the samples were compared with the normal control (see Figure 2). Surface expression of GPI-APs in monocytes of the patient and his mother were normal for CD157, while FLAER and CD14 showed very slightly decreased expression when compared to the control sample.

### 3.4. X Chromosome Inactivation Studies

The X-chromosome-inactivation study conducted on the maternal blood cells showed that the chromosomes were inactivated in a 67:33 ratio; that is, the X chromosomes showed a random inactivation pattern. It is worth noting that this ratio was similar to that obtained in the quantification of neutrophil subpopulations according to CD16 expression in flow cytometry (74:26). On the other hand, after enrichment of neutrophil subpopulations (with normal or decreased CD16 expression), the X-inactivation patterns were highly skewed (greater than 90%) for one or the other X chromosome, respectively, as expected for the inactivation of the variant or the wild-type allele of the *PIGA* gene (see Figure 1).

## 4. Discussion

Somatic variants in the *PIGA* gene are a well-known cause of PNH, but germline *PIGA* variants were thought to be lethal because of the absence of reported variants and results in animal models [16]. In 2012, Johnston and cols. reported the first family in which a nonsense germline *PIGA* variant was detected and caused a lethal X-linked phenotype. Later, more cases were reported and a phenotypic spectrum was described, ranging from a Fryns Syndrome-like phenotype to a purely neurological phenotype [17]. Our patient’s phenotype shared common characteristics with those previously reported: severe developmental delay (DD), early-onset refractory epilepsy, chorea, and dysmorphic features [4,5,17,18,19]. In contrast, no alkaline phosphatase elevation, alteration of iron metabolism or malformations of the internal organs were found. Regarding prognosis, an increased mortality rate has been associated with *PIGA* glycosylation disorder compared to other biosynthetic defects of the GPI anchors. In a reported case series, half of the patients died within the first two years of life [20], as in our case. 

Due to the small number of reported cases, no clear genotype–phenotype correlations could be established. However, Bayat and cols. [17] pointed out that variants affecting the p.R77 site with different amino-acid substitutions result in generalized seizures, profound intellectual disability (ID) and DD, and the onset of epilepsy around six months of age. These characteristics are also representative of our case. However, to date, there have been no reported cases with variants in the same position to allow phenotype comparisons.

Given the deleterious effect of pathogenic variants, it was thought that asymptomatic female carriers would show preferential X-inactivation. In fact, whenever these studies have been reported, preferential inactivation has been found in carrier females [2,4,5,11]. Thus, it can be inferred that *PIGA* variants with random inactivation in females do not have a deleterious effect and that their pathogenic role must therefore be questioned. 

The impact of *PIGA* variants has been previously studied through functional studies. Johnston et al. [2] identified the pathogenic stop variant c.1234C>T (p.Arg412Ter) in the *PIGA* gene in two affected boys with MCAHS2. In vitro functional-expression studies in *PIGA*-null cell lines showed that the R412X mutant protein retained some residual activity, with partial restoration of GPI-anchored proteins, suggesting that a small amount of full-length protein was generated by read-through of the stop codon. The findings indicated that GPI anchors are important for normal development, particularly of the central nervous system. Subsequently, Kato et al. [19] identified four *PIGA* variants in probands showing early-onset epileptic encephalopathies. Flow cytometry of blood granulocytes from the patients demonstrated reduced expression of GPI-anchored proteins. Accordingly, we decided to study the surface expression of GPI-APs on granulocytes by flow cytometry as a phenotypic assay to investigate the pathogenic role of the *PIGA* variant. Although there is currently not a standardized flow cytometry method for the analysis of GPI-APs expression in hematopoietic cells from patients with MCAHS, previous studies carried out in neutrophils demonstrated a clear reduction in the expression of CD16 and sometimes a moderate decrease in the expression of FLAER and another GPI-AP [19,21,22,23]. We analyzed the expression of different GPI markers (CD14, CD16, CD24, CD157, and FLAER) in peripheral blood samples from the patient, his mother, and a healthy control. A clear reduction in the intensity of CD16 expression was observed in the patient’s total neutrophils and in 26% of his mother’s neutrophils. Of the rest of the GPI markers, only CD24 and FLAER presented a moderate reduction when the samples were compared with the control, while CD157 expression was nearly normal. 

The expression of CD16 allowed the separation of the maternal sample neutrophils with normal expression from those with significantly reduced expression, facilitating a subsequent molecular analysis of the X-inactivation pattern. The obtained results proved that the inactivation pattern was highly skewed in opposite directions in these two groups of selected cells, demonstrating that the antigenic phenotype correlates with the activation of one or the other X chromosome in the carrier mother. This fact supports the hypothesis that the *PIGA* variant c.130C>G [p.Pro44Ala] results in an altered GPI synthesis, so the variant was classified as pathogenic. 

In terms of genetic counseling, as this is an X-linked recessive condition, it was explained to the family that there was a 25% chance of future offspring being an affected male. Considering the severity of this condition, two preventive options were offered: a pre-implantation genetic diagnosis of embryos obtained by assisted reproduction techniques or noninvasive prenatal testing (to ascertain the sex) and prenatal diagnosis by means of a chorionic biopsy of a male fetus resulting from a natural conception. As for the family study, it was not possible to perform genetic testing on the maternal grandmother. However, carrier testing was offered to both of the mother’s sisters, one of whom accepted the offer and tested negative.

A putative limitation of this study could be that the variant c.130C>G in *PIGA* gene is in fact present in somatic mosaicism in the mother, resulting in the production of two subpopulations of neutrophils with different antigenic patterns. However, there is no evidence that the variant is mosaic in the mother. On the other hand, in the inactivation study performed on the enriched fraction of neutrophils with normal expression of CD16, that is, on those cells presumably not carrying the variant, the pattern of inactivation of the X chromosome would be expected to be random. However, the result clearly showed skewed inactivation of the second X chromosome. Furthermore, the family history suggests an X-linked condition.

In conclusion, MCAHS2 is an X-linked recessive inherited disease related to the synthesis of GPIs caused by *PIGA* variants. With this study, we demonstrate that preferential inactivation of the X chromosome is not necessarily found in all female carriers. We therefore consider that in cases of random X-chromosome inactivation in carriers of variants of uncertain significance, evaluation of CD16 expression on neutrophils should be considered to clarify the pathogenic role of *PIGA* variants.

## Figures and Tables

**Figure 1 genes-15-00802-f001:**
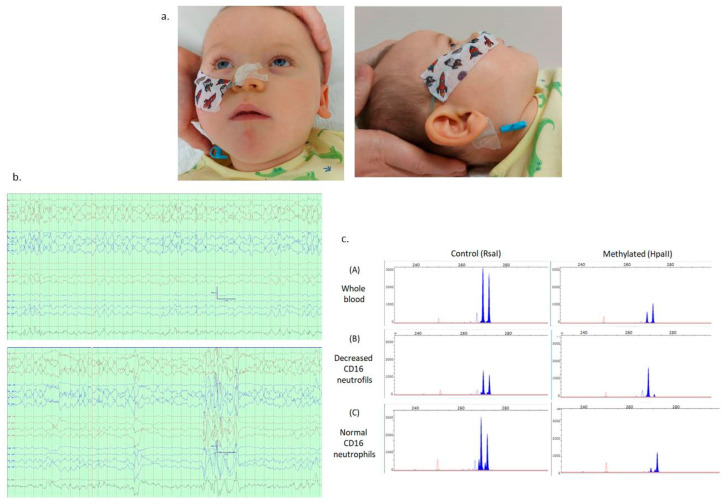
(**a**) Facial dysmorphic features include round and flat face; large, low-set and rotated ears; wide and depressed nasal bridge; and absence of tooth eruption. (**b**) Interictal EEG shows diffuse background slowing, frequent diffuse bursts of spike and slow waves, and posterior quadrant spikes and sharp waves. (**c**) X-chromosome-inactivation analysis of (**A**) whole blood cells, showing a random 67:33 ratio; (**B**) the neutrophil subpopulation enriched for decreased CD16 expression by flow cytometry; (**C**) the neutrophil subpopulation enriched for normal CD16 expression by flow cytometry.

**Figure 2 genes-15-00802-f002:**
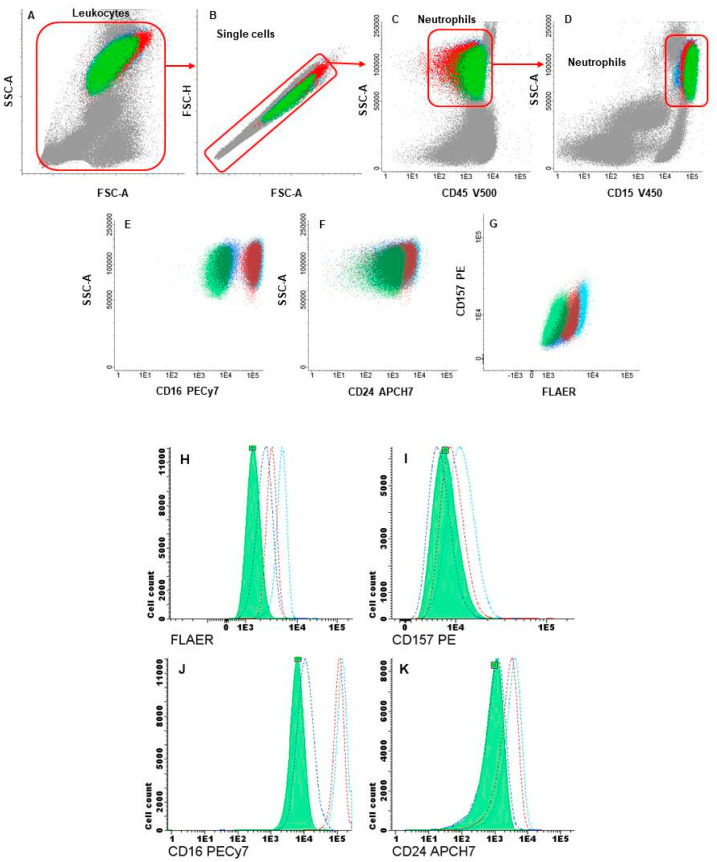
Flow-cytometry dot plots showing the gating strategy for the identification and characterization of the patient neutrophils (green), his mother’s normal neutrophils (light blue), his mother’s deficient neutrophils (dark blue), and healthy control neutrophils (red). Leukocytes and single cells were identified by FSC and SSC (**A**,**B**) and expression of CD45 (**C**) and CD15 (**D**); the characterization of the neutrophils was based in the analysis of the expression of CD16, CD24, CD157, and FLAER (**E**–**G**); squares indicate the mean fluorescence intensity for each neutrophil population. The histograms of FLAER, CD157, CD16, and CD24 expression (**H**–**K**) show the expression of these markers in the patient neutrophils (solid green), his mother’s normal neutrophils (light blue dashed line), his mother’s deficient neutrophils (dark blue dashed line), and healthy control neutrophils (red dashed line).

**Table 1 genes-15-00802-t001:** Cytometry and cell-sorting reagents.

	Reagent	Clone	Target Population
Cytometry, first step	FLAER Alexa Fluor 488a	NA	Neutrophils
			Monocytes
			(GPI marker)
	CD157 PEb	SY11B5	Neutrophils Monocytes (GPI protein)
	CD45 PercPb	2D1	Leukocytes
	CD15 HV450b	MMA	NeutrophiIs
	CD64 APCb	10.1	Monocytes
Cytometry, second step	CD16 PeCy7b (tube 1)	3G8	Neutrophils (GPI protein)
	CD24 APCH7b (tube 1)	ML5	Neutrophils (GPI protein)
	CD20 PerCPCy5.5b (tube 1)	L27	B lymphocytes
	CD14 APCH7b (tube 2)	MP9	Monocytes (GPI protein)
	CD24 PEc (tube 2)	ML5	Neutrophils (GPI protein)
Cell sorting	CD15 HV450b	MMA	Neutrophils
	CD45 HV500b	HI30	Leukocytes
	CD157 PEb	SY11B5	Neutrophils Monocytes (GPI protein)
	CD16 PECy7b	3G8	Neutrophils (GPI protein)
	CD20 PerCPCy5.5b	L27	B lymphocytes
	CD64 APCb	10.1	Monocytes
	FLAER Alexa Fluor 488a	NA	Neutrophils Monocytes (GPI marker)

## Data Availability

The original contributions presented in the study are included in the article, further inquiries can be directed to the corresponding author.

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
