# Peer review of "Multiple Congenital Anomalies-Hypotonia-Seizures Syndrome 2 Caused by a Novel PIGA Variant Not Associated with a Skewed X-Inactivation Pattern"

_genes, 2024, doi:10.3390/genes15060802_

Round 1

Reviewer 1 Report

Comments and Suggestions for Authors

General Comments:

This article reports a novel PIGA gene variant that causes Multiple Congenital Anomalies-Hypotonia-Seizures Syndrome 2 (MCAHS2) in a male infant. The variant was inherited from his mother, exhibiting a random X-chromosome inactivation pattern. The study employs whole-exome sequencing, Sanger sequencing, and flow cytometry for a detailed analysis, providing significant insights into the genetic and phenotypic characteristics of MCAHS2.

Strengths:

  1. Novelty: Identifies and describes a novel PIGA variant.
  2. Clinical Relevance: Provides valuable data for diagnosis and management.
  3. Methodological Rigor: Uses advanced techniques, yielding credible results.
  4. Detailed Analysis: Offers comprehensive analysis of the patient's phenotype and genetic data.
  5. Suggestions for Improvement:
  1. Functional Studies: Further literature review on the impact of PIGA variants.
  2. Broader Implications: Expand the discussion to explore implications for other X-linked genetic disorders.

Specific Recommendations:

  1. Introduction: Clarify the understanding of X-inactivation patterns.
  2. Materials and Methods: No comment.
  3. Results: Enhance the presentation of flow cytometry data.
  4. Discussion: Discuss potential limitations in more detail.
  5. Conclusion: Summarize the main findings concisely, emphasizing clinical applications. 

Author Response

Referee 1

Suggestions for improvement

Functional studies: Further literature review on the impact of PIGA variants.

As suggested, some lines regarding the impact of PIGA variants reported in the available literatura has been included in the discussion: “The impact of PIGA variants has been previously studied through functional studies.  Johnston et al. [2] identified the pathogenic stop variant c.1234C>T (p.Arg412Ter) in the PIGA gene in two affected boys with MCAHS2. In vitro functional expression studies in PIGA-null cell lines showed that the p.Arg412Ter mutant protein retained some residual activity with partial restoration of GPI-anchored proteins, suggesting that a small amount of full-length protein was generated by read-through of the stop codon. The findings indicated that GPI anchors are important for normal development, particularly of the central nervous system. Subsequently, Kato et al. [13] identified four PIGA mutations in probands showing early-onset epileptic encephalopathies. Flow cytometry of blood granulocytes from the patients demonstrated reduced expression of GPI-anchored proteins.”

Broader Implications: Expand the discussion to explore implications for other X-linked genetic disorders.

We believe it would be too speculative to suggest that the present study has implications for other X chromosome-related conditions. Furthermore, PIGA is the only early-stage GPI synthesis gene located on this chromosome. However, any suggestion on this sense would be welcome.

Specific recommendations

  • Introduction: clarify the understanding of the X-inactivation patterns.

A new paragraph has been included in the introduction as suggested: “Interestingly, the available literature has described a pattern of X-chromosome inactivation of almost 100% in mothers carrying the PIGA variant. The transcriptional silencing of one of the X chromosomes in XX individuals, also known as X-chromosome inactivation, is a dosage compensation strategy that occurs in most mammals. In placental mammals the prevalent form is random X-chromosome inactivation, so both paternal and maternal X chromosomes have roughly the same chance of being inactivated. However, preferential (skewed) inactivation of one X chromosome can occur by chance, but also due to a non-random choice, or as a result of selection for or against cells carrying one specific active or inactive X chromosome. This last mechanism occurs in individuals that carry X-linked variants associated with lethality or restricted survival and is a hallmark of carriers of a few X-linked diseases. There are several X-linked disorders for which all the pathogenic variants systematically associate a whole X-inactivation skewness, such as those in ATRX gene or for incontinentia pigmenti. In these cases, a preferential X-inactivation is pathognomonic and serves as a supporting functional assay. Other X-linked conditions are frequently, but not always, associated to a preferential X-inactivation, so that the analysis of the X-inactivation pattern may be suggestive, although a random X-inactivation pattern does not exclude their involvement. For the pathogenic variants in PIGA gene, it was though that asymptomatic female carriers would show preferential X-inactivation. In fact, whenever these studies have been re-ported, a preferential inactivation is found in carrier females. Thus, it could be inferred that PIGA variants with random inactivation in females do not have a deleterious effect and therefore their pathogenic role must be questioned.”

  • Results: Enhance the presentation of flow cytometry data.

A new figure to better represent the gating strategy for the identification and characterization of neutrophils is included.

  • Discussion: Discuss potential limitations in more detail.

A potential limitation was addressed and discussed as follows: “A putative limitation of this study could be that the variant c.130C>G in PIGA gene is in fact present in somatic mosaicism in the mother, causing two subpopulations of neutrophils with different antigenic patterns. However, there is no evidence that the variant is in mosaic in the mother. On the other hand, in the inactivation study performed in the enriched fraction of neutrophils with a normal expression of CD16, that is, those cells presumably not carrying the variant, the pattern of inactivation of the X chromosome would expected to be random. However, the result obtained was a clearly skewed inactivation of the second X chromosome. Furthermore, the family history suggests an X-linked condition”.

  • Conclusion: Summarize the main findings concisely, emphasizing clinical applications.

We have included a final conclusion paragraph aiming to briefly emphasize the relevant aspects as requested: “In conclusion, MCAHS2 is an X-linked recessive inherited disease related to the synthesis of GPIs caused by PIGA variants. With this study we demonstrate that preferential inactivation of the X chromosome is not necessarily found in all female carriers. We therefore consider that in cases of random X-chromosome inactivation in carriers of variants of uncertain significance, evaluation of CD16 expression on neutrophils should be considered to clarify the pathogenic role of PIGA variants.”

Reviewer 2 Report

Comments and Suggestions for Authors

This is an interesting paper to consider non-skewed X-inactivation in females carriers of PIGA variants.

The manuscript is well-written. However, I have some minor comments to the authors:

-Did they consider or explore any kind of mosaicism in the mother, how was the phenotype of the mother?

-Please, describe the genetic counseling for the family. In the manuscript, authors mentioned an affected cousin, they even mentioned a maternal aunt of the patient, but nothing regarding genetic counseling. I think this is relevant for the clinicians and the families.

-Lines 164 and 175. Please define abbreviations the first time you use them in the manuscript. ID/DD: Intellectual disability, and developmental delay?

Author Response

Referee 2

Comments

  • Did they consider or explore any kind of mosaicism in the mother, how was the phenotype of the mother?

The comment regarding the possibility of maternal mosaicism is now addressed in the 7th paragraph of the Discussion section.

New information regarding mother’s medical history has been added in the “clinical features” section: “No relevant phenotypic features were found in the mother and regarding her medical history, the problems reported so far were obesity, polycystic ovary syndrome and gestational diabetes”.

  • Please, describe the genetic counseling for the family. In the manuscript, authors mentioned an affected cousin, they even mentioned a maternal aunt of the patient, but nothing regarding genetic counseling. I think this is relevant for the clinicians and the families.

We have better explained and clarified the genetic counseling as requested by including a paragraph in the discussion (6th paragraph):  “In terms of genetic counseling, as this is an X-linked recessive condition, it was explained to the family that there was a 25% chance of having an affected male in their offspring. Considering the severity of this condition, two preventive options were offered, well to carry out a pre-implantation genetic diagnosis on embryos obtained by assisted reproduction techniques or to carry out a noninvasive prenatal testing (to ascertain the sex) and prenatal diagnosis by means of a chorionic biopsy of the male foetus resulting from a natural conception.  As for the family study, it was not possible to perform it on the maternal grandmother. However, carrier testing was offered to both of the mother's sisters, one of whom accepted and tested negative.”

  • Lines 164 and 175. Please define abbreviations the first time you use them in the manuscript. ID/DD: Intellectual disability, and developmental delay?

The definitions for those abbreviations have been included in the text in the above mentioned lines: “developmental delay (DD) // intellectual disability (ID)”.

Round 2

Reviewer 1 Report

Comments and Suggestions for Authors

The author has thoughtfully revised the article based on my questions.